# Physical Activity Levels in Leisure Time, Sociodemographic Characteristics, and Chronic Non-Communicable Diseases in Brazilian Older Adults: National Health Survey, 2019

**DOI:** 10.3390/ijerph20196887

**Published:** 2023-10-06

**Authors:** Ana Michele Saragozo de Freitas, Izabella Cristina da Silva dos Santos, Amanda Santos Da Silva, Ricardo Aurelio Carvalho Sampaio, Roberto Jerônimo dos Santos Silva

**Affiliations:** 1Programa de Pós-Graduação em Educação Física, Universidade Federal de Sergipe, São Cristóvão 49100-000, Brazil; anamichsaragozo@gmail.com (A.M.S.d.F.); sampaiorac@academico.ufs.br (R.A.C.S.); 2Núcleo de Estudos e Pesquisa em Aptidão Física, Saúde e Desempenho de Sergipe, NUPAFISE/UFS, Departamento de Educação Física, Universidade Federal de Sergipe, São Cristóvão 49100-000, Brazil; crisbela.1990@gmail.com (I.C.d.S.d.S.); amandaedf8@gmail.com (A.S.D.S.)

**Keywords:** aging, physical activity, chronic diseases

## Abstract

Health interventions for elderly people must understand the association between physical activity, sociodemographic factors, and non-communicable diseases. This study aimed to verify the association between physical activity in leisure time, sociodemographic factors, and NCDs in Brazilian older people. This is a descriptive study, with a cross-sectional design, carried out using secondary data from the 2019 National Health Survey—Brazil. It used data from 23,144 subjects aged over 60 years. Statistical analyses included descriptive and quantile regression with cutoff points 0.25, 0.50 (Median), 0.75, and 0.90 to verify the association between the variables. The statistical software R version 4.1.3 was used. Older people aged between 60 and 79 years were highlighted up to the 50th quantile. Females showed lower values in all quantiles, with emphasis from quantile 50 onwards. Subjects who self-declared as “white” showed significant differences up to quantile 50, not indicating significant values from this cutoff point. Residents of the rural area had lower values than residents of the urban area in all quantiles, with emphasis starting from the median. For cardiovascular diseases, it was found that subjects who did not report having this type of disease had better results for the amount of physical activity during leisure time, especially from the 75th quantile. It was concluded that there is a direct relationship between time spent in leisure-time performing physical activity, sociodemographic variables, and NCDs. It is necessary to review and validate cutoff points according to each category viewed, which can favor the adjustment of interventions according to each population. Actions of this nature can favor adherence by groups of older people to the weekly practice of physical activity.

## 1. Introduction

Brazil has shown considerable changes in demographic projections with a gradual increase in life expectancy [1,2]. This is a result of the decrease in mortality and fertility rates, resulting in greater aging of the population [1,2]. This situation can be verified by observing data from the Institute of Applied Economic Research—IPEA/Brazil, an agency that identifies economic and social variations in Brazil, which indicates that the proportion of older people in 2010 was 7.3% and projects a prevalence of 40.3% for 2100, generating an inversion in the demographic pyramid of the Brazilian population [1].

Aging is a natural, gradual, and heterogeneous process, determined by several factors such as genetics, lifestyle, living conditions, health conditions, and environmental conditions [3]. This process causes changes in body composition such as increased fat and a loss of muscle mass, resulting in reduced muscle strength and functional dependence [4]. Such changes lead to modifications in the lifestyle of the elderly, as they make it difficult to carry out daily activities independently [3].

Changes in lifestyle due to aging can favor the emergence or worsening of chronic non-communicable diseases (NCDs) [2,5], resulting in a reduced quality of life for these individuals. In Brazil, NCDs were responsible for about 54.7% of deaths recorded in 2019 and 11.5% through their diseases, of which the most frequent were cardiovascular diseases, neoplasms, diabetes, and chronic respiratory diseases [2].

Regarding the growth of NCD records, in Brazil, there is a higher prevalence of modifiable risk factors such as inadequate diet, smoking, obesity, alcohol, physical inactivity, and social determinants [2,6]. The low level of physical activity in older adults is related to all-cause mortality, especially NCDs [5,7]. It considerably increases the risk of diseases such as type 2 diabetes, coronary heart disease, and cancer, thus decreasing life expectancy [8]. Modifiable risk factors can be altered by behavioral changes and government actions through regulations, incentives, and reduced exposure to harmful health products [2].

Specifically, in Brazil, Ministerial Programs were presented focusing on the mapping, control and prevention of these diseases, which ranged from mapping policies such as the NHS (base document for this work) to intervention programs such as the “Family Health Strategy Program” and the “Health Academy Program”, which aim at community intervention and community care, including the practice of physical activity [2,3,4]. Thus, in 2013 and 2019, the Brazilian Ministry of Health, in partnership with the Brazilian Institute of Geography and Statistics (IBGE), carried out the National Health Survey/Brazil (NHS 2019) to reveal a broad health diagnosis of the Brazilian population, agreeing information on the determinants and conditions in health [9]. According to NHS 2019, NCDs are a public health problem for Brazil, which generates declines in quality of life [6] and an increase in functional disabilities, resulting in high economic costs for both society and health systems [10].

From all the changes resulting from the demographic transition, such as the population growth of older adults, increase in NCDs, and significant changes in the lifestyle of these individuals, there is the possibility of using other strategies to monitor health, living conditions, and diseases and their risk factors. In this context, the literature indicates that there is an association between NCDs and the level of leisure-time physical activity of Brazilians; however, there is a scientific gap regarding the specific group, as most studies do not analyze elderly people with some type of NCD or with some functional limitation and the average-based analysis is used in the vast majority of cases, leaving those with higher and lower levels out of its analysis. Therefore, it is interesting to understand the behavior of leisure-time physical activity according to certain quantiles, as these represent population behaviors and trends, suggesting a more detailed and attentive view of the variable’s behavior and minimizing biases due to population cutoff and adjustment. Therefore, this study aimed to explore the association between sociodemographic variables and NCDs regarding the level of leisure-time physical activity in Brazilian old people.

## 2. Materials and Methods

A descriptive cross-sectional study was conducted using secondary data from the NHS 2019 (https://www.ibge.gov.br/estatisticas/downloads-estatisticas.html?caminho=PNS/2019/Microdados/Dados, accessed on 7 August 2023). The research sought to determine health indicators in Brazilian territory through a population-based sample and a three-stage conglomerate sampling plan [1].

The total sample of NHS 2019 was 91,683 individuals, of which 23,144 (25.24%) were selected for the present study, considering as inclusion criteria people aged ≥ 60 years of both sexes, involving all Brazilian states; it should be noted that Brazilian legislation indicates that the subject aged ≥ 60 years is characterized as an older person [9,11].

To carry out the data collection of the NHS 2019, the sample considered originated from a reference survey composed of a set of units called primary (UPAs), secondary (households), and third parties (choice of a resident aged 18 years or older). Data collection occurred through a simple random sampling, involving trained coordinators, supervisors, and interviewers, using mobile devices [9].

It was carried out between August 2019 and March 2020, covering data from 1600 Brazilian municipalities through an interview. NHS 2019 was carried out by resolution No. 466/2012 of the National Health Council—NHC/Brazil and approved by the National Research Ethics Commission—CONEP/Brazil, under recommendation No. 328.159—2013 and No. 3.529.376—2019; its data are available for public access and use [10].

The questionnaire used by NHS 2019 was defined through interviews carried out in person, with questions about the population’s health conditions, access to and use of health services, information regarding morbidities, lifestyle, and dimension of exposure of the Brazilian population, among other topics. More information about the sampling plan, instruments, and other procedures used can be obtained through the official report of the research [10].

The procedures for data analysis used elements of descriptive statistics such as measures of central tendency (mean and median), dispersion (standard deviation), absolute frequencies, and categorical variables in frequency and percentiles. Quantile regression was used to analyze the association between the variables, taking as reference the main reference quantiles (25th, 50th, and 75th), as well as the 90th quantile, to present an overview of the trend of distribution and a closer look at the risk observed by the study [12]. These quantiles were chosen to analyze whether or not there are variations in association at each point, and these models were adjusted according to the distribution of the dependent variable, adopting a significance level of 5%. The analyses were performed using the statistical software R version 4.1.3 (10 March 2022)—“One Push-Up”.

The independent variables that characterized NCDs were inserted into the model hierarchically. The questions that characterized arterial hypertension, diabetes mellitus, and heart diseases considered the following questions: Q2a—“Has any doctor ever given you the diagnosis of arterial hypertension?”; Q30a—“Has any doctor ever given you a diagnosis of diabetes?”; Q63a—“Has any doctor ever diagnosed you with a heart disease, such as a heart attack, angina, heart failure or other?” [2].

For the verification of leisure-time physical activity, the sample was analyzed according to the weekly volume, considering with greater volume the subjects who practiced about 150 min of weekly physical activity, according to the classification given by the World Health Organization (WHO) [5]. Thus, the variable was characterized by the questions: P34—“Did you practice any type of physical exercise and/or sport?”; P35—“How many days a week do you usually practice physical exercise or sport?”; P37—“In general, on the day you practice (practiced) exercise or sport, how long in minutes does this activity last?”; P36—“What physical exercise or sport did you practice (practiced) most often?”; “Other, please specify (10) ____________.” More detailed information regarding the study variables, as well as their characterizations and categorizations, can be obtained in Table 1.

## 3. Results

The sample resulted in 23,144 older people, with a mean of 69.99 ± 7.86 years old, whose largest portion (86.7%) was people between 60 and 79 years old, classified as “older adults”; 12,740 (55.04%) were female, 13,071 (56.5%) declared race/ethnicity of non-white skin, 17,689 (76.4%) pointed out to know how to read and write, and 17,673 (76.4%) had housing in an urban area. These figures are presented in Table 2.

Table 1 shows the characterization of the sample based on the responses obtained through the data collection instrument and the categorization made by IBGE [10], accompanied by the confidence intervals of the sociodemographic variables selected according to the objectives of the study: “age group” (“older adults and oldest-old adults”), “gender” (“male and female”), “race/ethnicity” (“white and non-white”), “education” (“can read and write and cannot read and write”), and “housing” (“urban and rural”). The figures presented below depict the behavior of leisure-time physical activity, when compared to sociodemographic variables, indicating “1A”: behavior of leisure-time physical activity time about the age group; “1B”: behavior of leisure-time physical activity about biological gender; “1C”: behavior of leisure-time physical activity about self-perception of race/ethnicity; “1D”: behavior of leisure-time physical activity about education; “1E”: behavior of leisure-time physical activity about housing (Figure 1A–E). Upon reviewing Figure 1 in general, it is verified that regression lines with higher correlation have steeper slopes, indicating that the lower the physical activity, the smaller the difference between the groups/quantiles in each variable.

When analyzing Figure 1A, it can be observed that the “older adults” from 60 to 79 years old stand out with a higher level of physical activity in leisure time compared to the “oldest-old adults” (80 years or more); such a difference begins to stand out on a larger scale from the 50th quantile (t = 22.29, *p* < 0.001) going up to the 90th quantile (t = 12.81, *p* < 0.001). The same happens with the variables “education” and “housing” (Figure 1D,E), that is, subjects with a higher level of education and who live in an urban area have a higher level of physical activity in leisure time, highlighting the 50th quantile (t = 23.42, *p* < 0.001/t = 26.84, *p* = <0.001).

In relation to the variables “gender” and “race/ethnicity” (Figure 1B,C), it is clear that the variable “race/ethnicity” has significance between the groups in quantiles 25 (t = 23.32, *p* = <0.000) and 50 (t = 14.19, *p* = <0.000). In the “gender” variable, there is a greater behavior difference between the groups from quantile 75 onwards, with greater significance in quantile 90 (t = 6.22, *p* = <0.000), indicating that males have a higher level of physical activity during leisure time compared to the females.

Table 3 shows the health conditions of the individuals in the sample, represented by the variables that characterize chronic non-communicable diseases (NCDs) as “hypertension”, “diabetes mellitus”, and “not specified heart diseases”, resulting in a higher frequency of hypertension (55%), followed by diabetes mellitus (20%) and finally, those affected by heart diseases (12%).

Figure 2 shows the association between leisure-time physical activity and NCDs, identifying “2A”: behavior of leisure-time physical activity in relation to hypertension; “2B”: behavior of leisure-time physical activity in relation to diabetes mellitus; “2C”: behavior of leisure-time physical activity in relation to unspecified heart disease (Figure 2A–C).

Individuals who reported not having “hypertension”, “diabetes mellitus”, and “not specified heart diseases” had a higher level of leisure-time physical activity compared to those who had these diseases. However, when analyzing the quantiles, “arterial hypertension” showed greater behavior than subjects’ physical activity levels after quantile 75 and in quantile 90 (t = 4.87, *p* = <0.001), even greater behavior, whereas those who did not report “arterial hypertension” had a higher level of leisure-time physical activity than those who have this condition (Figure 2A).

For “diabetes mellitus” and “not specified heart diseases”, the inclination line starts from the median (t = 23.24, *p* < 0.001 and t = 16.307, *p* < 0.001), meaning there is greater influence with a lesser amount of physical activity in leisure time (Figure 2B,C, respectively); this trend continues until quantile 75 (t = 8.36, *p* < 0.001 and t = 5.57, *p* < 0.001).

Figure 3 shows the frequency of independent variables (NCDs) in relation to the level of leisure-time physical activity according to each category of NCDs, indicating that in the group of those who reported not having any NCDs, there is greater practice of physical activity during leisure time, as already mentioned above. Both groups have the same median value, observing differences from the 50 quantiles (Figure 3).

It can also be seen that the largest volume of time spent practicing physical activity during leisure time in the groups with some type of chronic disease is much lower than the largest volume of time in the opposite group, especially in the groups that reported having “diabetes mellitus” and “not specified heart diseases”.

## 4. Discussion

The present study explored the associations between sociodemographic variables and NCDs with physical activity and leisure time among Brazilian older people through secondary data provided by NHS 2019. It revealed that in different quantiles, men classified as “older adults” (aged between 60 and 79 years), with some degree of education and who lived in an urban area, declared themselves more active in leisure time. When analyzing those who reported having some type of NCD, it was found that they were less active compared to those who did not report any NCD and that older people who reported “hypertension” were more active compared to those who were diagnosed with “diabetes mellitus” and “not specified heart diseases”.

Economic and sociodemographic changes, together with the link with the health of Brazilians, especially those defined by inequalities, have been the focus of discussions, as well as health promotion and prevention models and the relationship with their determinants [13,14], assisting in a more accurate assessment of the performance of physical activity among a given population [15]. Associated with several NCDs and premature mortality, the insufficiency of physical activity is responsible not only for a negative impact on the mental health and quality of life of the subjects but is also responsible for high economic expenses [16]. Among the NCDs, the ones that stand out the most are cardiovascular diseases, which are currently considered the largest cause of death in Brazil, with their prevalence increasing according to the aging of the population [17].

According to WHO, NCDs and their risk factors are strongly related to gender, race/ethnicity, level of education, occupation, and income [18]. Marmot and Bell found evidence in a whole systems approach that, especially in underdeveloped countries, NCDs have an increased risk rate when people’s economic level is lower [19]. The factor “being female” along with older age is associated with lower functional capacity [3,20], where older women have a poorer quality of life despite living longer compared to the opposite sex [21]. Considering that there are many differences between men and women in their different occupational roles, experiences, and opportunities in society, this may justify one of the conclusions of our study, showing that men are more active than women. These data are also consistent with the NHS 2013, with higher percentages found for men than women at 27.3% and 18.6%, respectively. Data were collected from all individuals in the sample, not just the elderly. It is clear that there is a need to know the basis of more active aging modulators in relation to biological sex [10,21].

Another important factor that we found in our study that has been corroborated by other authors [3,13,22] is the difference between age groups affecting the level of leisure-time physical activity. Elderly people who were older were more likely to present severe functional dependence, which can be explained not only by the aging process, but also by the high prevalence of NCDs with increasing age; this is related to the findings of this study, as the oldest-old people achieved a lower level of physical activity in leisure time compared to the elderly (between 60 and 79 years old), mainly in the 50th quantile.

According to the NHS 2019, as age advances, the level of leisure-time physical activity tends to decrease, and in the age group of 60 years or more, there is a low percentage of elderly people who performed the 150 min of leisure-time physical activity recommended by WHO, that being only 19.8% [10,18]. Considering that in one of the studies the three domains of physical activity (leisure, work, and commuting to work) were analyzed, more than half of the subjects aged 60 years or older were classified as insufficiently active [10], results similar to the findings of the present study, analyzed only within the domain of leisure, as previously presented.

Another important point found in our study was that people with some type of education have a higher level of physical activity during leisure time, compared to those who do not, because, as the following authors [13,23] point out, people with low education have a higher prevalence of NCDs, possibly due to less access to protective factors, such as health services, food, and physical activity. The same occurs when individuals with low education levels are in locations that are difficult to access and less developed, as they are affected by NCDs as a result of greater vulnerability [24]. The education factor, when positive, is also linked to a better condition of life, a greater understanding of the benefits of a more active lifestyle, as well as better opportunities (employment, salary, and purchasing power), and greater contact with environments that favor the practice of physical activity [25]. In NHS 2019, it was disclosed that the level of physical activity in leisure time increased with the level of education [10].

In relation to “race/ethnicity”, Melo et al., as well as Simões et al., show that the prevalence of NCDs is generally linked to economic factors and the environment in which the subject is inserted [13,23]. The NHS 2019 [10] itself reveals that people considered “white” have a higher level of physical activity, considering all domains (leisure, work, commuting, and domestic activities); these data differ from our study, which considered only the leisure domain, presenting that in the most active elderly people there is no difference between races/ethnicities (Figure 1D).

In the analyses related to NCDs, it was observed that those who had a higher level of physical activity during leisure time reported not having any NCDs. A multihospital trial carried out by Doukky et al. through the analysis of data from the Heart Failure Adherence and Retention Trial (HART) [26] showed that individuals with cardiovascular conditions spend on average about 14 times longer performing sedentary behavior when compared to the time spent walking in leisure time. In a systematic review [27], publications on adherence to physical exercise in chronic patients and older people were analyzed, and some of the factors related to adherence were identified, namely: initial evaluations of barriers and facilitators for practice, design of the exercise program, social support, self-efficacy and integration into daily life, in addition to the presence of a multidisciplinary team, and the education of participants, among other factors.

In other words, NCD risk factors such as physical activity can be modified through prevention strategies and measures [18]. The monitoring of NCDs and their prevalence is fundamental for the creation of plans and the improvement of health policies [10], such as increasing levels of physical activity (e.g., about 150 min a week of physical activity with mild or moderate intensity or 75 min with vigorous intensity) to help in the fight and treatment of NCDs [5].

The present study has limitations regarding self-reporting on the investigated variables, and there may be misconceptions about memories, as well as in the language and schooling of the respondents. However, this tool is widely used for epidemiological research, enabling reliable results [25]. Another limitation was the lack of information on physical activity reports provided by elderly people with some type of disability. Furthermore, having three times as many elderly people living in urban areas may have masked certain findings or magnitudes of results. Regarding the strengths, we have the use of a national sample, adequately quantified and represented, with a significant sample size, in addition to the analysis used in the study. Quantile analysis is little used in health-related research, but it is robust and provides a complete overview of the relationship between the variables, enabling new conclusions about the data since it can be analyzed at different distribution points or through the median, without the interference of outliers [28]. In addition, the present study will contribute to the development of strategies for the promotion of leisure-time physical activity for older Brazilians, taking into account socio-demographic and cultural diversities.

## 5. Conclusions

Thus, it is concluded that there are differences in behavior between the variables analyzed, with men and elderly people aged 60 to 79 years, with some degree of education, who live in an urban area and without any NCD, are more active in leisure time. Within the group of those who have some NCD, those who reported having “high blood pressure” are more active compared to those who have been diagnosed with “diabetes mellitus” and “not specified heart diseases”.

These findings reinforce the need for strategies to promote active aging adapted to and appropriate for men and women of different age groups to avoid the permanence and extension of gender inequalities [20], as well as sociodemographic inequalities. The findings of this study also reinforce the importance of knowledge of barriers and facilitators for the practice of physical activity in leisure time for Brazilian older people.

Therefore, taking action and monitoring NCDs are of great importance, and greater investigations should be considered regarding the theme of oldest-old adults, especially into modifiable factors, such as physical activity, and thus more specific work should be implemented with the individuals mentioned.

## Figures and Tables

**Figure 1 ijerph-20-06887-f001:**
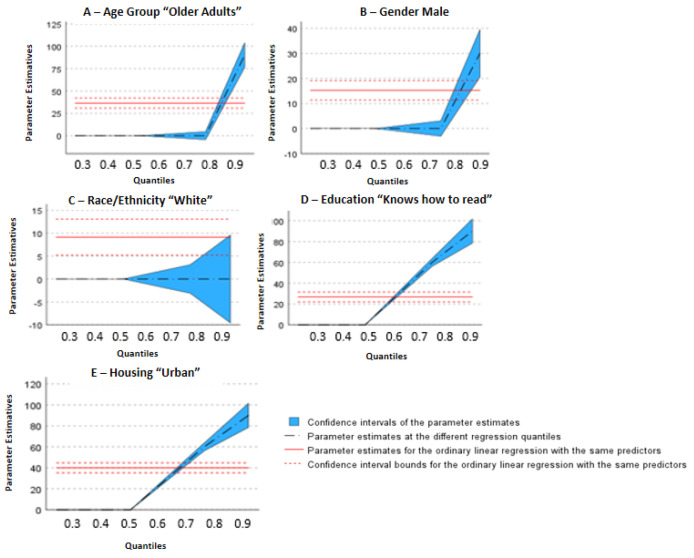
(**A**–**E**) Association between sociodemographic variables and leisure-time physical activity considering the 25th, 50th, 75th, and 90th percentiles.

**Figure 2 ijerph-20-06887-f002:**
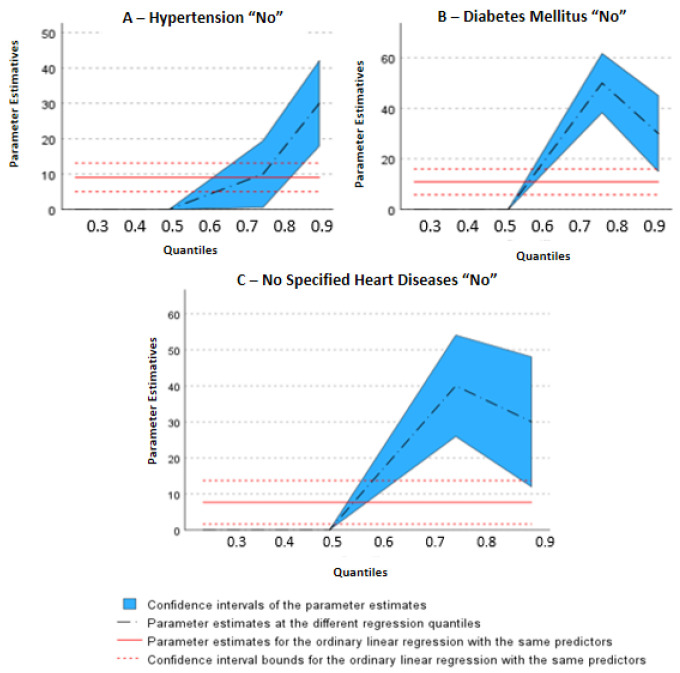
(**A**–**C**) Graphical representation of quantile regression relating chronic non-communicable diseases with physical activity performed during leisure time.

**Figure 3 ijerph-20-06887-f003:**
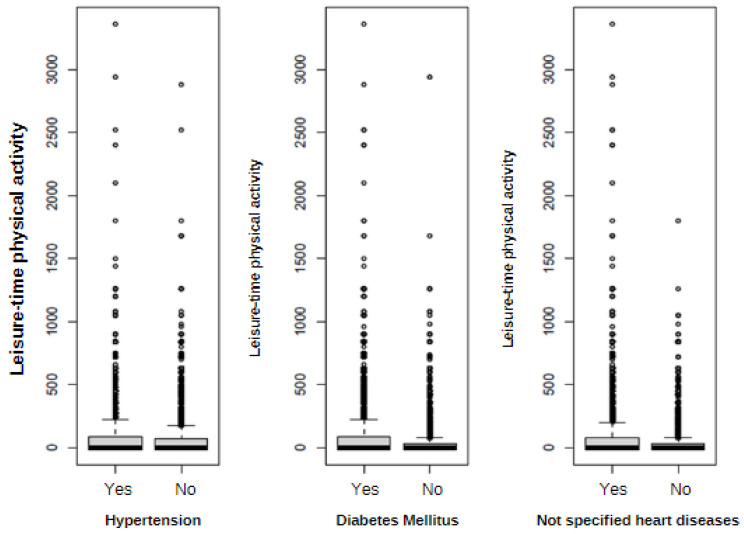
Graphical representation of leisure-time physical activity levels in each NCD considered in the study.

**Table 1 ijerph-20-06887-t001:** Presentation, characterization, and categorization of the variables used in the study.

Variable Type	Question *	Variable	Variable Characteristic	Categorization
**DEPENDENT**	Sum of questions: P034 P035 P036 P03701 P03702 P03601	Leisure-Time Physical Activity	Continuous Nominal—dichotomous WHO classification was adopted [5]	in Minutes/Week (0) Highest volume weekly (≥150 min) (1) Lowest weekly volume (≤150 min)
**INDEPENDENT**	C008	Age Group	Nominal dichotomized (using ≥60 years as cutting bridge for the characterization of “elderly”; using the classification in “long-lived-elderly” for those aged 80 years or over). Possibility of greater autonomy [3]	The classification was adopted: (0) Older adults 60–79 years (1) Oldest-old adults 80–107 years
	C006	Gender	Biological	(0) Male (1) Female
	C009	Race/ethnicity	Nominal—dichotomized (definition presented in the NHS document)	(0) White (1) Non-white
	D00901	Education	Nominal—dichotomized (definition presented in the NHS document)	(0) Knows how to read. (1) Cannot read
	V0026	Housing	Nominal—dichotomized (definition presented in the NHS document)	(0) Urban (1) Rural
	Q00201	Hypertension	Nominal—dichotomized	(0) No (1) Yes
	Q03001	Diabetes Mellitus	Nominal—dichotomized	(0) No (1) Yes
	Q06306	Not specified heart diseases	Nominal—dichotomized	(0) No (1) Yes

* Questions taken from the questionnaire used to collect population data; it is available through this link: https://www.pns.icict.fiocruz.br/questionarios/ (accessed on 6 March 2023).

**Table 2 ijerph-20-06887-t002:** Sample characterization—NHS 2019 data.

Sociodemographic Variables		*n*	%	% (CI 95%)	
				Lower Limit	Upper Limit
Age Group	Older adults	20,057	86.7	86.2	87.1
	Oldest-old adults	3087	13.3	12.9	13.8
Gender	Male	10,404	44.95	44.3	45.6
	Female	12,740	55.04	54.4	55.7
Race/ethnicity	White	10,071	43.51	42.9	44.2
	Non-white	13,071	56.48	55.8	57.1
Education	Reads and writes	17,689	76.43	75.9	77.0
	Cannot read or write	5455	23.56	23.0	24.1
Housing	Urban	17,673	76.36	75.8	76.9
	Rural	5471	23.63	23.1	24.2

CI: confidence interval; lower limit; upper limit.

**Table 3 ijerph-20-06887-t003:** Health conditions of the sample individuals—NHS 2019 data.

Variable		*n*		CI 95%	
			%	Lower Limit	Upper Limit
Hypertension	No	10,142	45	44.3	45.6
	Yes	12,428	55	54.4	55.7
Diabetes Mellitus	No	17,722	80	79.9	81.0
	Yes	4305	20	19.0	20.1
Not specified heart diseases	No	20,004	88	87.6	88.4
	Yes	2724	12	11.6	12.4

CI: confidence interval; lower limit; upper limit.

## Data Availability

Data from this research can be obtained from https://www.ibge.gov.br/estatisticas/downloads-estatisticas.html?caminho=PNS/2019/Microdados/Dados (accessed on 1 September 2023).

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
