# Peer review of "Physical Activity Levels in Leisure Time, Sociodemographic Characteristics, and Chronic Non-Communicable Diseases in Brazilian Older Adults: National Health Survey, 2019"

_ijerph, 2023, doi:10.3390/ijerph20196887_

Round 1
Reviewer 1 Report
Reviewer Report: ijerph-2617419
Title: Physical activity level in leisure time, sociodemographic characteristics and chronic non-communicable diseases in Brazilian older adults: National Health Survey, 2019
Dear authors,
The study in question is relevant for understanding the association between leisure-time physical activity, sociodemographic characteristics, and non-communicable chronic diseases (NCDs) in Brazilian older adults. The involvement of a sample of 23,144 individuals aged 60 and older and the multidimensional approach demonstrate the strength of the study. Additionally, the results suggest that promoting physical activity can be beneficial for the health of older adults, especially those who do not report cardiovascular diseases. Nevertheless, I have identified some areas that require improvement, such as the introduction, and especially the discussion section, which needs to be more in-depth, particularly in light of the study's findings. Below are my specific comments for enhancing the work.
Best regards,
The reviewer
---------------------------------------------------------------------------------------------------------------------------
Abstract:
- Please provide context for the study, explaining the importance of the relationship between physical activity, sociodemographic factors, and non-communicable diseases (NCDs) in Brazilian older adults.
- The following sentence is confusing and needs further adjustment: "There was a trend towards an increase in the time of physical activity in leisure time from the median."
- Please revise the conclusion to address the study's objectives and point out future implications.
Introduction:
- Recently, the Brazilian Institute of Statistics published the updated population census. Could the authors indicate any real growth trend in the Brazilian elderly population, rather than just projected growth?
- If the authors are referring to adopting an unhealthy lifestyle, please clarify in lines 43-44.
- The authors mention a scientific gap concerning the specific group in this study and the different quantiles of physical activity levels. However, they do not indicate the relevance of these physical activity quintiles in the presented context. In this regard, the authors may want to state the importance of this clearly.
Methods:
- Please indicate the location (e.g., a link) where the data can be found.
- A quick internet search found that Brazil has more than 5,000 cities. Is the sample representative of all cities, both large and small?
- Although it was mentioned that a cluster sampling plan was used, could you explain a bit more about how the primary, secondary, and tertiary sampling units were selected?
- Please move lines 98 to 107 after Table 1.
Results and Discussion:
- Are the confidence intervals correct? For example, "Older adults 86.7" should be within the confidence interval. Also, some lines in the % column do not sum up to 100%.
- The first sentence of the discussion is lengthy and confusing (lines 207-212). Could it be divided for better clarity? Furthermore, what are the implications of these results for public health? Authors can improve the start of the discussion by addressing these points.
- The second paragraph is confusing and seems disconnected from the study's objectives. Please revise the second paragraph to improve the clarity of ideas.
- Lines 217 and 219 are missing brackets [13] and [14]. See 19 in line 232.
- The factors for which differences were observed have been overlooked. Please highlight the main findings of the study based on the objectives and discuss possible explanations.
- Please include limitations related to the cross-sectional design, and the absence of information about physical activity reporting provided by older adults (e.g., for the older adults, or those with impairments, who provided the activity reports?). Additionally, having three times more elderly residents in urban areas might have masked certain findings or result magnitudes.
- Please verify the references and confirm whether the journal titles should be italicized or not, following the author's guidelines. The year of publication should be in bold.
Author Response
In reference to manuscript: ijerph-2617419
Title: Physical activity level in leisure time, sociodemographic characteristics and chronic non-communicable diseases in Brazilian older adults: National Health Survey, 2019
REVIEWER #1
Abstract:
- Please provide context for the study, explaining the importance of the relationship between physical activity, sociodemographic factors, and non-communicable diseases (NCDs) in Brazilian older adults.
Response: Health intervention for elderly people must understand the association between physical activity, sociodemographic factors and non-communicable diseases.
- The following sentence is confusing and needs further adjustment: "There was a trend towards an increase in the time of physical activity in leisure time from the median."
Response: Dear reviewer, the sentence has been changed for better understanding.
- Please revise the conclusion to address the study's objectives and point out future implications.
Response: Dear reviewer, the text has been adjusted.
Introduction:
- Recently, the Brazilian Institute of Statistics published the updated population census. Could the authors indicate any real growth trend in the Brazilian elderly population, rather than just projected growth?
Response: Dear reviewer, the latest CENSO publication (year 2022) does not specify the conditions of elderly people in Brazil, perhaps because it is a preliminary publication, as https://censo2022.ibge.gov.br/panorama/downloads.html?localidade=BR or https://biblioteca.ibge.gov.br/visualizacao/livros/liv102011.pdf
- If the authors are referring to adopting an unhealthy lifestyle, please clarify in lines 43-44.
Response: The sentence in lines 43-44 has been adjusted to improve understanding of the text.
- The authors mention a scientific gap concerning the specific group in this study and the different quantiles of physical activity levels. However, they do not indicate the relevance of these physical activity quintiles in the presented context. In this regard, the authors may want to state the importance of this clearly.
Response: The quantiles used present the population behavior and trend, suggesting a more detailed and attentive view of the variable's behavior, minimizing biases due to population cutoff.
Methods:
- Please indicate the location (e.g., a link) where the data can be found.
Response: Data can be found at https://www.ibge.gov.br/estatisticas/downloads-estatisticas.html?caminho=PNS/2019/Microdados/Dados
- A quick internet search found that Brazil has more than 5,000 cities. Is the sample representative of all cities, both large and small?
Response: This information can be obtained in reference nine, as indicated in the text or at https://www.pns.icict.fiocruz.br/delineamento-da-pns/
- Although it was mentioned that a cluster sampling plan was used, could you explain a bit more about how the primary, secondary, and tertiary sampling units were selected?
Response: This information can be obtained in reference nine, as indicated in the text or at https://www.pns.icict.fiocruz.br/delineamento-da-pns/
- Please move lines 98 to 107 after Table 1.
Response: In order to satisfy the reviewer and have a better aesthetic organization of the text, we thought it would be better to move lines 102 to 128 after table 1.
Results and Discussion:
- Are the confidence intervals correct? For example, "Older adults 86.7" should be within the confidence interval. Also, some lines in the % column do not sum up to 100%.
Response: Adjustments were made as requested by the reviewer.
- The first sentence of the discussion is lengthy and confusing (lines 207-212). Could it be divided for better clarity? Furthermore, what are the implications of these results for public health? Authors can improve the start of the discussion by addressing these points.
Response: Dear reviewer, this information is present in the following paragraph.
- The second paragraph is confusing and seems disconnected from the study's objectives. Please revise the second paragraph to improve the clarity of ideas.
Response: As requested by the reviewer, adjustments were made.
- Lines 217 and 219 are missing brackets [13] and [14]. See 19 in line 232.
Response: As requested by the reviewer, adjustments were made.
- The factors for which differences were observed have been overlooked. Please highlight the main findings of the study based on the objectives and discuss possible explanations.
Response: As requested by the reviewer, adjustments were made.
- Please include limitations related to the cross-sectional design, and the absence of information about physical activity reporting provided by older adults (e.g., for the older adults, or those with impairments, who provided the activity reports?). Additionally, having three times more elderly residents in urban areas might have masked certain findings or result magnitudes.
Response: Dear reviewer, for what the study proposes, the cross-sectional design is appropriate, since there was no intention to define cause and effect! The limitation added in the text, thanks for posting!
- Please verify the references and confirm whether the journal titles should be italicized or not, following the author's guidelines. The year of publication should be in bold.
Response: Verified! References were added according to the model provided by the IJERPH.

Reviewer 2 Report
This is a descriptive investigation into a subject of undeniable interest, but one that has already been studied extensively. Nevertheless, it gives us an interesting portrait of a population.
Overall, the article is well written. The methodology, discussion and conclusions are well thought out and well-founded.
The introduction could be improved, especially in terms of the relationship between physical activity level in leisure time and the prevention of chronic non-communicable diseases.
Author Response
In reference to manuscript: ijerph-2617419
Title: Physical activity level in leisure time, sociodemographic characteristics and chronic non-communicable diseases in Brazilian older adults: National Health Survey, 2019
REVIEWER #2
This is a descriptive investigation into a subject of undeniable interest, but one that has already been studied extensively. Nevertheless, it gives us an interesting portrait of a population.
Overall, the article is well written. The methodology, discussion and conclusions are well thought out and well-founded.
The introduction could be improved, especially in terms of the relationship between physical activity level in leisure time and the prevention of chronic non-communicable diseases.
Dear Reviewer!
Thank you for your consideration and compliments on our manuscript!
The introduction and other topics of the manuscript were adjusted and corrected.
Best regards,

Reviewer 3 Report
This study aims to examine associations of sociodemographic variables and NCDs with leisure-time physical activity in Brazilian older people using quantile regression analysis. The manuscript has several issues that need to be addressed.
1. Abstract. The conclusion in the abstract needs to be rewritten. The current one is perspective, but not a conclusion,
2. The motivation of this study is not clear. There is insufficient information to reach your research question in the introduction section. Future readers cannot understand the gap between this study and previous studies. Therefore, future readers also cannot judge the rationale, strength, and significance of this study. Please provide a further explanation.
3. Materials and Methods. Please provide the reference for the validity of the questions regarding physical activity assessment.
4. Figures. Although the labels in the figures indicate that the blue color is the confidence intervals, no such information is presented in the figures.
5. Line 186: “For “diabetes mellitus” and “not” is a misrepresentation. Also, lines 194 to 202 are hard to follow. Please modify.
6. Conclusions. Line 299: The statement “men in the age group of 60 to 79 year … are more active in leisure” is kind of confusing, because there are no comparisons among the age groups stratified by sex.
Author Response
In reference to manuscript: ijerph-2617419
Title: Physical activity level in leisure time, sociodemographic characteristics and chronic non-communicable diseases in Brazilian older adults: National Health Survey, 2019
REVIEWER #3
- Abstract. The conclusion in the abstract needs to be rewritten. The current one is perspective, but not a conclusion.
Response: According to the reviewer's instructions, the text was revised and adjusted.
- The motivation of this study is not clear. There is insufficient information to reach your research question in the introduction section. Future readers cannot understand the gap between this study and previous studies. Therefore, future readers also cannot judge the rationale, strength, and significance of this study. Please provide a further explanation.
Response: Dear reviewer, it is interesting to understand the behavior of leisure-time physical activity according to certain quantiles since these present the behavior and population trend, suggesting a more detailed and attentive view of the behavior of the variable, minimizing biases due to population cut-off. The text has been checked and adjusted, thank you for the comments.
- Materials and Methods. Please provide the reference for the validity of the questions regarding physical activity assessment.
Response: Dear reviewer, the link to check the validity of the questions about physical activity is >https://www.pns.icict.fiocruz.br/questionarios/< but this information can also be found in reference 9.
- Figures. Although the labels in the figures indicate that the blue color is the confidence intervals, no such information is presented in the figures.
Response: Dear reviewer, I believe it is not possible to view the confidence intervals due to the short range they present, which can be confirmed by checking table 1.
- Line 186: “For “diabetes mellitus” and “not” is a misrepresentation. Also, lines 194 to 202 are hard to follow. Please modify.
Response: The text was modified according to the reviewer's instructions, thank you for the comments.
- Conclusions. Line 299: The statement “men in the age group of 60 to 79 year … are more active in leisure” is kind of confusing, because there are no comparisons among the age groups stratified by sex.
Response: Some changes were made to the text for a better understanding of the conclusion of the study.

Reviewer 4 Report
From the perspective of the subject matter, I can say that the research is interesting and relevant because it focuses on the relationship between leisure-time physical activity, sociodemographic characteristics, and non-communicable chronic diseases (NCDs) among the elderly population in Brazil. The study's results highlight the need to develop active aging promotion strategies tailored to different age and gender groups and emphasize the importance of monitoring and investigating NCDs among the elderly.
However, I believe the paper has some shortcomings, which I would like to address further:
1. In the introduction chapter, very few details are provided about previous strategies. It might be helpful to list more of them and provide a critical analysis in terms of leisure-time physical activity in general and for subjects with NCDs in particular.
2. In the Materials and Methods chapter, it should be specified how the questionnaire used in NHS 2019 was administered. Was it conducted in-person, online, or as a hybrid approach?
3. The Discussion chapter is not well-structured. Many statements made by the study's authors refer to bibliographic sources without comparing the data from those studies with the data obtained from their own study to convince readers that the data are similar. Another limitation could be that leisure-time physical activities, based on physical exercise as a form of movement, should be separated from other physical activities such as domestic chores, which serve a different purpose, although they indirectly contribute to health prevention. These are distinct activities.
Overall, the paper is fairly well-crafted, but from my perspective, the major disadvantage is the presentation of previous strategies and the concrete comparison of the data obtained in the current study with data from other studies.
Author Response
In reference to manuscript: ijerph-2617419
Title: Physical activity level in leisure time, sociodemographic characteristics and chronic non-communicable diseases in Brazilian older adults: National Health Survey, 2019
REVIEWER #4
- In the introduction chapter, very few details are provided about previous strategies. It might be helpful to list more of them and provide a critical analysis in terms of leisure-time physical activity in general and for subjects with NCDs in particular.
Response: I would like to thank the reviewer for his comments and say that we did not touch on this point directly because I believe it is an element that is widely discussed in the literature, however, the text underwent some adjustments that favor this understanding.
- In the Materials and Methods chapter, it should be specified how the questionnaire used in NHS 2019 was administered. Was it conducted in-person, online, or as a hybrid approach?
Response: The text was adjusted according to the reviewer’s suggestion.
- The Discussion chapter is not well-structured. Many statements made by the study's authors refer to bibliographic sources without comparing the data from those studies with the data obtained from their own study to convince readers that the data are similar. Another limitation could be that leisure-time physical activities, based on physical exercise as a form of movement, should be separated from other physical activities such as domestic chores, which serve a different purpose, although they indirectly contribute to health prevention. These are distinct activities.
Response: Dear reviewer, this division was considered in the study, as the HNS already addresses the topic from this perspective!
Overall, the paper is fairly well-crafted, but from my perspective, the major disadvantage is the presentation of previous strategies and the concrete comparison of the data obtained in the current study with data from other studies.
Response: Taking into account all the suggestions made here by the reviewer, some adjustments were made to the text.

Round 2
Reviewer 1 Report
Dear Authors,
After resubmitting the manuscript, I was able to observe that most of the recommendations were adjusted. In this sense, I forward my report to the Editor-in-Chief.
Best regards,
Author Response
Dear reviewer,
Thank you for all the suggestions and for approving the correction of the manuscript!
Regards.
Reviewer 3 Report
Thank you for your responses. The reviewer has seen the authors' effort. I have the follow-up comments.
1. The motivation of this study remains unclear. The explanation of using quantile regression is somewhat vague. Please clarify this issue.
2. Regarding the confidence intervals in the figures, please provide the statistics (parameter estimates and the 95% confidence intervals) in tables (e.g., in the supplemental material) if it is not possible to view the confidence intervals due to the short range they present. The 95% CI in Table 1 gives no such information.
3. Line 200, “For “diabetes mellitus” and “and “not specified heart diseases”, Remove the 2nd “and” in this sentence.
4. Line 244-245, “It was revealing that, in different quantiles, revealing that men”. Please correct this sentence.
Author Response
Dear reviewer,
Please see the attached file!
Regards

Reviewer 4 Report
Dear authors,
Yes, the paper has improved significantly, but I still insist on the following aspects:
1. In the Introduction chapter, I return to the same issue regarding previous policies and strategies. I'm referring to the ones mentioned in lines 61-63 and in references 2-4. Even though the specialized literature is vast in this direction, I believe it's essential to specify a few key aspects of the strategies you're referring to, as they apply specifically to your country (Brazil). Perhaps other countries, more or less developed, have implemented different strategies with more or less good results, but I think it's important to be aware of these aspects.
2. Additionally, in the Discussion section, I believe it would be beneficial to present concrete data from your study (percentages, etc.) in comparison to data obtained from other similar studies.
Kind regards
Author Response

(The authors gave the same response as above.)
